# Sorafenib Inhibits Ribonucleotide Reductase Regulatory Subunit M2 (RRM2) in Hepatocellular Carcinoma Cells

**DOI:** 10.3390/biom10010117

**Published:** 2020-01-09

**Authors:** Pei-Ming Yang, Li-Shan Lin, Tsang-Pai Liu

**Affiliations:** 1Graduate Institute of Cancer Biology and Drug Discovery, College of Medical Science and Technology, Taipei Medical University, Taipei 11031, Taiwan; yangpm@tmu.edu.tw; 2PhD Program for Cancer Molecular Biology and Drug Discovery, College of Medical Science and Technology, Taipei Medical University and Academia Sinica, Taipei 11031, Taiwan; 3TMU Research Center of Cancer Translational Medicine, Taipei 11031, Taiwan; 4Cancer Center, Wan Fang Hospital, Taipei Medical University, Taipei 11696, Taiwan; 5Department of Surgery, Mackay Memorial Hospital, Taipei 10449, Taiwan; 6Mackay Junior College of Medicine, Nursing and Management, New Taipei City 11260, Taiwan; 7Department of Medicine, Mackay Medical College, New Taipei City 25245, Taiwan; 8Liver Medical Center, Mackay Memorial Hospital, Taipei 10449, Taiwan

**Keywords:** autophagy, hepatocellular carcinoma, ribonucleotide reductase, sorafenib

## Abstract

The main curative treatments for hepatocellular carcinoma (HCC) are surgical resection and liver transplantation, which only benefits 15% to 25% of patients. In addition, HCC is highly refractory and resistant to cytotoxic chemotherapy. Although several multi-kinase inhibitors, such as sorafenib, regorafenib, and lenvatinib, have been approved for treating advanced HCC, only a short increase of median overall survival in HCC patients was achieved. Therefore, there is an urgent need to design more effective strategies for advanced HCC patients. Human ribonucleotide reductase is responsible for the conversion of ribonucleoside diphosphate to 2′-deoxyribonucleoside diphosphate to maintain the homeostasis of nucleotide pools. In this study, mining the cancer genomics and proteomics data revealed that ribonucleotide reductase regulatory subunit M2 (RRM2) serves as a prognosis biomarker and a therapeutic target for HCC. The RNA sequencing (RNA-Seq) analysis and public microarray data mining found that RRM2 was a novel molecular target of sorafenib in HCC cells. In vitro experiments validated that sorafenib inhibits RRM2 expression in HCC cells, which is positively associated with the anticancer activity of sorafenib. Although both RRM2 knockdown and sorafenib induced autophagy in HCC cells, restoration of RRM2 expression did not rescue HCC cells from sorafenib-induced autophagy and growth inhibition. However, long-term colony formation assay indicated that RRM2 overexpression partially rescues HCC cells from the cytotoxicity of sorafenib. Therefore, this study identifies that RRM2 is a novel target of sorafenib, partially contributing to its anticancer activity in HCC cells.

## 1. Introduction

According to the global cancer statistics of 2018 [1], liver cancer belongs to the sixth most common cancer and is the fourth leading cause of cancer death around the world. Hepatocellular carcinoma (HCC) accounts for the 75–85% of primary liver cancer cases [1]. The major risk factors for HCC consist of chronic infection with hepatitis B virus (HBV) and hepatitis C virus (HCV), dietary toxin exposure (such as aflatoxin and aristolochic acid), metabolic diseases (such as fatty liver disease and diabetes), and alcohol addiction [2]. A multi-step histological process is involved in the neoplastic evolution of HCC. Hepatic injury induces necrosis, followed by hepatocyte proliferation. Continuous destructive-regenerative cycles develop a chronic liver disease, leading to liver cirrhosis that is characterized by fibrosis and abnormal nodule formation. Then, hyperplastic and dysplastic nodules are observed and, ultimately, HCC develops. HCC is further categorized into well-, moderately-, and poorly differentiated tumors, of which the last one represents the most malignant form of primary HCC [3].

The management of HCC includes surgical resection, liver transplantation, percutaneous local ablations (such as ethanol injection, radiofrequency thermal ablation), transarterial chemoembolization (TACE), transarterial radioembolization (TARE), and systemic pharmacological treatments [2,4]. The major curative treatments are surgical resection and liver transplantation, which, however, are only suitable for 15% to 25% of patients [5]. Actually, 70% of patients relapse after resection [6]. Multi-kinase and immune-checkpoint inhibitors have been regarded as new treatment options for advanced HCC. The multi-kinase inhibitor sorafenib was the first systemic molecular-targeted therapy approved for advanced HCC by the United States (US) Food and Drug Administration (FDA, Washington DC, USA) in 2007 and it improved the median overall survival from 8 to 11 months [7]. Ten years later, in 2017 and 2018, other multi-kinase inhibitors (regorafenib, lenvatinib, and cabozantinib) were also approved [8,9,10]. Ramucirumab, a monoclonal antibody targeting the vascular endothelial growth factor receptor 2 (VEGFR2), was approved in 2019 [11]. Two human anti-PD-1 monoclonal antibodies, nivolumab and pembrolizumab, were approved in 2017 and 2018, respectively [12,13].

Sorafenib is an orally-available multi-kinase inhibitor. RAF/MEK (mitogen-activated protein kinase/ERK kinase)/ERK (extracellular signal-regulated kinase) cascade, VEGFR, and platelet-derived growth factor receptor (PDGFR) are viewed as the primary targets for its anti-proliferative and anti-angiogenic effects [14,15]. Downregulation of the components of cell survival pathways, such as myeloid cell leukemia-1 (MCL-1), and upregulation of pro-apoptotic Bcl-2 family proteins, such as p53 upregulated modulator of apoptosis (PUMA) and Bcl-2–interacting mediator of cell death (BIM), are associated with the apoptosis-inducing effects of sorafenib in HCC [14,15]. Recently, several novel anticancer targets of sorafenib have been identified, such as 5-hydroxytryptamine (5-HT) receptor [16], stearoyl coenzyme A desaturase 1 (SCD1) [17], and protein biosynthesis [18]. Especially, sorafenib is identified as a system x_c_^−^ (a cell-surface Na^+^-independent cystine–glutamate antiporter) inhibitor that leads to endoplasmic reticulum (ER) stress and ferroptosis (an iron-dependent form of non-apoptotic programmed cell death) [19]. Sorafenib can also induce autophagy (a cytosolic process that delivers cytoplasmic constituents to the lysosome for degradation) in HCC. However, the role of autophagy in its anticancer activity is still controversial [20].

Human ribonucleotide reductase (RR) is a heterotetrameric complex by two large subunits RRM1 (ribonucleotide reductase catalytic subunit M1) and two small subunits RRM2 (ribonucleotide reductase regulatory subunit M2). RR maintains the homeostasis of nucleotide pools by converting ribonucleoside diphosphate to 2′-deoxyribonucleoside diphosphate [21]. Increased RR expression and activity are associated with malignant transformation and growth [22]. The critical role of RR in DNA synthesis and repair promotes it as an important anticancer target [23]. For example, an anti-RRM2 siRNA duplex exhibits anti-proliferative activity in cancer cells [24,25,26]. Small-molecule RR inhibitors (such as gemcitabine and hydroxyurea), either as single agents or in combination with other therapies, are shown to be a useful strategy for treating solid tumors and hematological malignancies [23]. A nucleoside analog, gemcitabine, is known to inactivate RR activity through binding to RRM1 by its diphosphate form [27]. Gemcitabine has been approved by the FDA for treating various cancers, including breast, non-small cell lung, ovarian, and pancreatic cancers. The RRM2 can be substituted by p53-inducible small subunit RRM2B (ribonucleotide reductase regulatory TP53 inducible subunit M2B; also known as p53R2) that regulates the synthesis of deoxyribonucleotide triphosphates (dNTPs) required for DNA damage repair [28]. Unlike RRM1 and RRM2, p53R2 suppresses invasiveness and its expression is usually associated with a better prognosis for cancer patients [29,30].

Sorafenib is the first approved molecular-targeted therapy for advanced HCC in a first-line setting since 2007 [7]. However, sorafenib benefits in only about 30% of patients, and acquired resistance often develops within six months [7,31,32]. In addition, sorafenib and other recent approved multi-kinase inhibitors only provide a short increase of median overall survival in HCC patients [7,8,9,11,32]. More understanding of the mechanisms of action of these multi-kinase inhibitors may identify valuable therapeutic targets in HCC, thereby facilitating the development of a more effective anticancer strategy. Here, we identify that inhibition of RRM2 expression is a common effect of sorafenib, which partially contributes to its anticancer activity in HCC cells. Our results suggest a novel mechanism of action of sorafenib for treating cancers.

## 2. Materials and Methods

### 2.1. Chemicals and Reagents

Dulbecco’s modified Eagle medium (DMEM), l-glutamine, sodium pyruvate, and antibiotic-antimycotic solution (penicillin G, streptomycin and amphotericin B), Lipofectamine 3000 transfection reagent, RNAiMAX transfection reagent, M-PER mammalian protein extraction reagent, and TRIzol reagent were purchased from ThermoFisher Scientific (San Jose, CA, USA). Fetal bovine serum was purchased from GIBCO (Grand Island, NY, USA). RT^2^ First Strand kit and RT² SYBR Green ROX qPCR Mastermix were purchased from Qiagen (Valencia, CA, USA). RRM2, Actin, and GAPDH antibodies were purchased from GeneTex (Hsinchu, Taiwan). The LC3B antibody was purchased from Cell Signaling Technology (Beverly, MA, USA). Horseradish peroxidase (HRP)-labeled secondary antibodies were purchased from Jackson ImmunoResearch Laboratories (West Grove, PA, USA). Sorafenib p-toluenesulfonate salt (sorafenib tosylate) was purchased from LC Laboratories (Woburn, MA, USA). Chloroquine, MG132, Giemsa stain solution, 3-(4,5-Dimethyl-2-thiazolyl)-2,5-diphenyl-2H-tetrazolium bromide (MTT), and dimethyl sulfoxide (DMSO) were purchased from Sigma (St. Louis, MO, USA). Human RRM2-encoding plasmid (pcDNA3-RRM2) and its control vector (pcDNA3) were purchased from Addgene (Watertown, MA, USA). ON-TARGETplus RRM2 siRNA, and non-targeting siRNA pool were purchased from Dharmacon (Lafayette, CO, USA).

### 2.2. Cell Culture and Treatment

HepG2 and PLC/PRF/5 (PLC5) cells were purchased from the Bioresource Collection and Research Center (Hsinchu, Taiwan). Cells were cultured at 37 °C in DMEM supplemented with 10% fetal bovine serum (FBS), 1% l-glutamine, 1 mM sodium pyruvate, 1% antibiotic-antimycotic solution, and incubated in a humidified incubator containing 5% CO_2_. According to the cell proliferation test (please see the “Results” section for details), sorafenib at 5 μM was used for most experiments. Because sorafenib tosylate is the generic ingredient in Nexavar (the brand name of sorafenib), it was used in this study.

### 2.3. RNA Sequencing (RNA-Seq)

Total RNA was isolated from sorafenib (5 μM for 24 h)-treated and untreated HepG2 and PLC5 cells by the TRIzol reagent, according to the manufacturer’s instructions. Sequencing and data analysis (two biological replicates) were performed using the BGISEQ-500 platform (Beijing Genomics Institute, Beijing, China). Briefly, the target RNA was selected by oligo (dT) magnetic beads for mRNA with polyA tail, and then fragmented and reversely transcribed to double-strand complementary DNA (dscDNA) by N6 random primers. The ends of dscDNA were repaired with phosphate at the 5′ end and A at the 3′ end for adaptor ligation. After polymerase chain reaction (PCR) amplification, the PCR products were denatured by heat and the single-stranded DNA was cyclized by splint oligo and DNA ligase. Then, the prepared library was sequenced [33,34]. On average, 23,655,455 raw sequencing reads were generated. After filtering low quality, 23,950,466 clean reads were mapped to reference using the HISAT/Bowtie2 tool [35,36] with the average mapping ratio of 80.17%. Gene expression level was quantified by a software package called RSEM [37]. Differentially expressed genes (DEGs) were identified using the NOISeq method following the default criteria: |fold-change| ≥ 2 and diverge probability ≥ 0.8 [38].

### 2.4. Microarray Analysis

The microarray results of sorafenib (3 μM for 24 h)-treated Hep3B and Huh7 cells (GSE96796 [39]) were obtained from the Gene Expression Omnibus (GEO) at the National Center for Biotechnology Information (NCBI). DEGs were identified using the R-based online tool, GEO2R [40], according to the following criteria: |Log_2_ fold-change| > 1 and *p*-value < 0.01.

### 2.5. Bioinformatics Analyses

The VENNY 2.1 web tool (https://bioinfogp.cnb.csic.es/tools/venny/) was used to generate a Venn diagram to visualize the overlapped genes. The gene expression profiling interactive analysis (GEPIA; http://gepia.cancer-pku.cn/ [41]) was used to analyze the mRNA levels of follistatin/FST and RRM2 genes in normal liver and cancer tissues and their impacts on cancer patients’ overall survival in the Liver Hepatocellular Carcinoma (LIHC) data set of The Cancer Genome Atlas (TCGA). The cBioPortal website (http://www.cbioportal.org/ [42,43]) was used to analyze the genetic alterations of the RRM2 gene in hepatocellular carcinoma (TCGA, PanCancer Atlas—LIHC data set). The Oncomine database (http://www.oncomine.org/ [44]) was used to examine the RRM2 mRNA expression in normal liver and cancer tissues. The Human Protein Atlas (HPA; http://www.proteinatlas.org/ [45,46]) was used to explore the RRM2 protein expression in normal and cancer tissues. The National Cancer Institute (NCI, Bethesda, MD, USA) Transcriptional Pharmacodynamics Workbench (https://tpwb.nci.nih.gov/Gene ExpressionNCI60/TPWorkbench/ [47]) was used to explore the effect of sorafenib on RRM2 mRNA expression in an NCI-60 cancer cell panel.

### 2.6. Real-Time Quantitative Polymerase Chain Reaction (qPCR)

Total RNA was isolated from sorafenib (5 μM for 24 and 48 h)-treated and untreated HepG2 and PLC5 cells by the TRIzol reagent, according to the manufacturer’s instructions. The first-strand cDNA was synthesized using the RT^2^ First Strand kit, following by PCR amplification using the RT² SYBR Green ROX qPCR Mastermix on a LightCycler real-time PCR system (Roche, Indianapolis, IN, USA) in triplicates. The fold-changes in gene expression were calculated using the comparative cycle threshold method. The used primer pairs were as follows: human RRM2 (forward 5′-GCGATTTAGCCAAGAAGTTCAGAT-3′ and reverse 5′-CCCAGTCTGCCTTCTTCTTGA-3′) [48] and human 18S ribosomal (r)RNA (forward 5′-CGGCGACGACCCATTCGAAC-3′ and reverse 5′-GAATCGAACCCTGATTCCCCGTC-3′) [49].

### 2.7. Western Blot Analysis

Total cell lysates were prepared using the M-PER mammalian protein extraction reagent. After 30 min incubation on ice and then centrifugation at 13,000× *g* for 20 min at 4 °C, the supernatant was collected and protein concentration was measured by the Bio-Rad protein assay. Equal amounts of protein (50 μg) were separated in 7%~12% sodium dodecyl sulfate (SDS)-polyacrylamide gel and then transferred to nitrocellulose membrane. The membrane was hybridized with the specific primary antibody at 4 °C overnight. After washing, the membrane was hybridized with a horseradish peroxidase-conjugated secondary antibody for 30 min at room temperature. The immunoblots were visualized by the enhanced chemiluminescence (ECL) reagent.

### 2.8. Cell Proliferation Assay

Cell proliferation was examined based on the incorporation of thymidine analog bromodeoxyuridine (BrdU) into newly synthesized DNA using the BrdU cell proliferation assay kit (BioVision, Mountain View, CA, USA). Briefly, cells (1 × 10^4^ for HepG2 cells and 7.5 × 10^3^ for PLC5 cells) were spread in 96-well plate and cultured overnight. After drug treatment for 48 h, cells were incubated with BrdU solution at 37 °C for 2 h, followed by 30 min incubation at room temperature (RT) in fixing/denaturing solution. Then, cells were incubated with BrdU detection antibody solution at RT for 1 h with gentle shaking. After washing twice, anti-mouse HRP-linked antibody solution was added to each well and the plate was placed at RT for 1 h. After washing 3 times, 3,3′,5,5′-tetramethylbenzidine (TMB) substrate was added to each well and the absorbance at 450 nm was measured after color development.

### 2.9. Colony Formation Assay

Long-term cell viability was determined by colony formation assay. Briefly, cells (5 × 10^3^) were cultured in 6-well plates and treated with drugs for 48 h. After washing with phosphate-buffered saline (PBS) twice, cells were cultured for 10 to 14 days. The colonies were stained with Giemsa stain solution.

### 2.10. Transient Transfection

The human RRM2 and control siRNAs were reversely transfected into cells by RNAiMAX transfection reagent. The human RRM2-overexpressing plasmid (pCMV-RRM2) and its control vector (pCMV) were transfected into cells by Lipofectamine 3000 transfection reagent. After 24~48 h, transfected cells were used for experiments.

### 2.11. Statistical Analysis

Statistical analysis was performed by the built-in programs in each database used in this study.

## 3. Results

### 3.1. RNA Sequencing (RNA-Seq) Identifies Ribonucleotide Reductase Regulatory Subunit M2 (RRM2) as a Novel Target of Sorafenib in HCC

Sorafenib was the first approved multi-kinase inhibitor for HCC. However, sorafenib benefits only 30% of HCC patients and the acquired resistance usually develops within six months [7,32]. Thus, the understanding of the mechanisms of action of sorafenib may help to design strategies to potentiate its limited antitumor activity. To investigate the potential mechanisms of action of sorafenib in HCC, HepG2 and PLC5 cells were treated with 5 μM sorafenib for 24 h, and then total RNAs were analyzed by the RNA-Seq. The DEGs induced by sorafenib in these cells are listed in Appendix A. In addition, microarray data for sorafenib-treated Huh7 and Hep3B (GSE96796 [39]) were obtained from the GEO database at the NCBI [40]. The DEGs in sorafenib-treated Huh7 and Hep3B cells are also listed in Appendix A. We found that two genes (FST and RRM2) were commonly downregulated by sorafenib in four HCC cell lines (Figure 1A). To investigate the roles of FST and RRM2 genes in HCC, we analyzed the TCGA-LIHC data set via the GEPIA web-based tool [41]. The results found that mRNA expression of RRM2, but not FST, was significantly higher in cancer tissues than that in normal tissues (Figure 1B). In addition, the mRNA level of RRM2 was increased during the cancer progression from stage I to III (Figure 1C), suggesting that RRM2 plays a role in the tumorigenesis of HCC. Furthermore, HCC patients with higher RRM2 mRNA expression had a lower overall survival (Figure 1D). Therefore, we propose that the downregulation of RRM2 by sorafenib may provide clinical benefits.

### 3.2. Frequent Overexpression of RRM2 in HCC

To explore the role of RRM2 in HCC, the copy number alterations and mutation status of RRM2 were compared with its mRNA expression by analyzing the “TCGA, PanCancer Atlas—LIHC” data set via the cBioPortal website [42,43]. As shown in Figure 2A, no mutation of the RRM2 gene was found. The copy number of the RRM2 gene was amplified in 9 (2.6%) out of 348 cases while RRM2 mRNA was upregulated in 48 (13.8%) out of 348 cases, suggesting that copy number amplification did not fully account for the RRM2 upregulation in HCC. To confirm the RRM2 upregulation in HCC, RRM2 expression profiles were analyzed using the existing cDNA microarray data sets [50,51,52,53] in the Oncomine database [44]. As shown in Figure 2B, significant increases of RRM2 mRNA were found in HCC compared with normal liver tissues. To investigate whether RRM2 protein was also overexpressed, the immunohistochemical (IHC) staining of RRM2 protein in normal and cancer tissues was obtained from the HPA database [45,46]. As shown in Figure 2C, 2 out of 10 cases (20%) have high/medium RRM2 staining, which is the highest among 20 cancer types (Figure 2D). In contrast, the expression of RRM2 protein was undetectable in normal liver tissues. These results suggest that RRM2 is overexpressed in HCC and maybe an attractive molecular target for treating HCC.

### 3.3. In Vitro Experiments Validate that Sorafenib Inhibits RRM2 mRNA and Protein Expressions

To confirm whether sorafenib indeed inhibits RRM2 expression, HepG2 and PLC5 cells were treated with sorafenib for 24 and 48 h, and then real-time qPCR and Western blot analyses were performed to examined the mRNA and protein expressions, respectively. As shown in Figure 3A, treatment with 5 μM sorafenib for 24 and 48 h inhibited RRM2 mRNA expression. In addition, 5 and 10 μM sorafenib significantly inhibited RRM2 protein expression at 24 and 48 h (Figure 3B). To further confirm the ability of sorafenib to downregulate RRM2 mRNA, we mined their relationship in an NCI-60 cancer cell panel via the NCI Transcriptional Pharmacodynamics Workbench [47]. We found that 5 and 10 μM sorafenib inhibited RRM2 mRNA expression at 24 h in various cancer cells (Figure 3C), suggesting that inhibition of RRM2 is a common effect of sorafenib in cancer cells. To investigate whether sorafenib also downregulated RRM2 via promoting its degradation, HepG2 and PLC5 cells were treated with sorafenib with or without a proteasome inhibitor MG132. Western blot analysis found that sorafenib-induced RRM2 downregulation was reversed by MG132. Therefore, sorafenib inhibits RRM2 expression by suppressing its transcription and promoting its protein degradation.

### 3.4. Inhibition of RRM2 Induces Autophagy in HCC Cells

Because RR is the rate-limiting enzyme for DNA synthesis [23], inhibition of RRM2 may interfere with DNA synthesis and cell proliferation. To confirm the role of RRM2 inhibition in the cell proliferation of HCC cells, a BrdU incorporation assay was performed. HepG2 and PLC5 cells were treated with RRM2 siRNA (si-RRM2) and its control scramble siRNA (si-Cont). Knockdown of RRM2 protein expression was confirmed by Western blot analysis (Figure 4C). Inhibition of RRM2 expression by siRNA significantly reduced cell proliferation (Figure 4A), which was comparable to the anticancer activity of sorafenib (Figure 4B). These results suggest the dependency of HCC cells on RRM2 expression.

Sorafenib is known to induce autophagy in cancer cells [20,54]. Because RRM2 inhibition has been reported to induce autophagy [55], we hypothesized that RRM2 inhibition may contribute to the autophagy-inducing effect of sorafenib. Indeed, RRM2 knockdown was sufficient to induce the accumulation of LC3B-II (Figure 4C, lanes 1, 2, 5, 6), an autophagic marker [56]. Because LC3B-II accumulation may result from the induction of autophagy or the reduction of autophagosome turnover [56], chloroquine was used to neutralize the lysosomal pH and prevent lysosomal degradation. If an autophagic flux is occurring, more accumulation of LC3B-II will be observed [56]. As shown in Figure 4C (lanes 3, 4, 7, 8), the amounts of LC3B-II were higher in the presence of chloroquine. To examine whether sorafenib induces autophagy in HCC cells, HepG2 and PLC5 cells were treated with sorafenib, with or without chloroquine, and then autophagy was detected by the accumulation of LC3B-II. As shown in Figure 4D, sorafenib induced LC3B-II accumulation after 48-h treatment, especially in the presence of chloroquine. To further confirm the sorafenib-induced autophagy, SQSTM1 (p62) protein turnover was analyzed by Western blotting. During autophagy, p62 and p62-bound polyubiquitinated proteins are degraded by lysosomes, thus serving as an indicator of autophagic flux [56]. Consistently, sorafenib inhibited p62 protein expression in a time-dependent manner (Figure 4E).

### 3.5. Overexpression of RRM2 Partially Rescues Sorafenib-induced Long-term Cytotoxicy in HCC Cells

To confirm the role of RRM2 inhibition in sorafenib-induced autophagy and cell death, HepG2 and PLC5 cells were transfected with RRM2-encoding plasmid (pcDNA3-RRM2) or its control vector (pcDNA3). Then, sorafenib-induced autophagy and cell death were examined by Western blot analysis and cell proliferation assay, respectively. For Western blot analysis, chloroquine was used to enhance the visualization of LC3B-II accumulation according to the result in Figure 4D. As shown in Figure 5A, RRM2 overexpression did not block sorafenib-induced LC3B-II accumulation in the presence of chloroquine. However, we found that the transfected RRM2 was suppressed by sorafenib, which may partially explain the inability of RRM2 overexpression to rescue sorafenib-induced autophagy. Similarly, RRM2 overexpression did not inhibit sorafenib-induced cell death (Figure 5B). To further examine the effect of RRM2 overexpression on sorafenib’s anticancer activity, a long-term colony formation assay was performed. Interestingly, we found that RRM2 overexpression could partially rescue cells from the growth-inhibitory effect of sorafenib (Figure 5C).

## 4. Discussion

Although sorafenib has opened a new window for treating HCC, the overall outcomes are still not satisfactory. In addition, the development of sorafenib resistance is common in HCC. Therefore, it is of great importance to explore the mechanisms of action of sorafenib to develop more effective individualized therapeutic strategies [14,15]. In this study, we identified that RRM2 upregulation was associated with poor prognosis and could be a novel therapeutic target for HCC. We also demonstrated that inhibition of RRM2 expression was a novel effect of sorafenib. Because RR has been recognized as a promising cancer therapeutic target and several small molecules, such as hydroxyurea and gemcitabine, are active in clinical uses [57], our finding may extend the clinical application of sorafenib.

Previous studies regarding the role of RRM2 in HCC are limited. It has been found that the HBV induces RRM2 expression via activating DNA damage response and targeting RRM2 by small molecules could inhibit HBV replication [58,59]. Similarly, RRM2 is also upregulated by the HCV and promotes viral RNA replication [60]. These studies suggest that RR- or RRM2-targeting agents may serve as potential antiviral agents for HBV/HCV infection and HBV/HCV-related HCC. Indeed, high RRM2 expression was observed in 210 of 259 (81.1%) HCC patients and correlated with their poor prognosis [61]. RRM2 has been considered a therapeutic target for HCC through a siRNA-based functional survey and bioinformatics analysis [62,63]. Delivery of RRM2 siRNA alone or in combination with adriamycin could inhibit xenografted or orthotopic tumor growth established in nude mice [62,64]. Supportively, our results showed that RRM2 siRNA transfection efficiently inhibited the cell proliferation of HCC cells.

Sorafenib was originally designed as an RAF kinase inhibitor, but it was later identified as a multi-kinase inhibitor. Inhibition of these targets, such as VEGFR and PDGFR, largely contributes to its clinical efficacy [65]. In recent years, other novel molecular targets of sorafenib have been discovered [16,17,18,19]. We previously mined the Connectivity Map (CMap) database to explore the potential molecular target(s) of sorafenib and found that sorafenib indirectly inhibited HDAC activity in HCC cells [66]. In this study, we further demonstrated that RRM2 is a common target of sorafenib, which minimally contributed to the anticancer activity of sorafenib in HCC. It is reasonable that ectopic RRM2 overexpression did not rescue HCC cells from the growth-inhibitory effect of sorafenib because RRM2 is only one of the multiple targets of sorafenib. In addition, we found that the ectopic RRM2 protein expression was efficiently reduced by sorafenib, which may also explain the observations in this study.

The regulation of RRM2 gene transcription is still largely unclear. Few transcription factors, such as E2F1 and E2F3, have been shown to activate RRM2 gene transcription in cancer cells [67,68]. However, our unpublished results indicated that overexpression of E2F1 did not restore the expression of RRM2 in response to sorafenib treatment in HCC cells. Tri-methylation of histone H3K36 (H3K36me3), an epigenetic mark associated with actively transcribed genes, is found to facilitate RRM2 transcription [69]. Because our previous study found that sorafenib exhibits HDAC-inhibitory activity [66], we proposed that inhibition of HDAC by sorafenib may lead to the accumulation of acetylated histone H3K36 (H3K36ac), thereby suppressing RRM2 transcription. Supportively, an HDAC inhibitor, mocetinostat, has been found to reduce RRM2 expression in human leiomyosarcoma cells [70].

Our results also indicated that sorafenib induced RRM2 proteasomal degradation, which cannot be rescued by chloroquine. Supportively, the previous study also shows that RRM2 undergoes proteasome-dependent degradation during autophagy [55]. Interestingly, we found that chloroquine alone could reduce RRM2 protein expression, which is similar to the effect of bafilomycin A1 (a lysosomal proton pump inhibitor that can raise the lysosomal pH) [55]. Therefore, inhibition of autophagic flux by chloroquine and bafilomycin A1 also downregulates RRM2. However, the involved mechanism has not been elucidated.

The F-box protein cyclin F is the major ubiquitin ligase that mediates the degradation of RRM2 through proteasome pathway, which can be activated by the phosphorylation of RRM2 at Thr33 by cyclin-dependent kinases 1/2 (CDK1/2) [71]. Whether sorafenib promotes RRM2 degradation through this pathway warrants further investigations. Recently, RRM2 acetylation has been shown to regulate RR activity [72]. Acetylation of RRM2 at Lys95 by lysine acetyltransferase 7 (KAT7) disrupts RRM2 homodimerization and inactivates RR. On the other hand, sirtuin 2 (SIRT2) deacetylases RRM2, leading to RR activation [72]. It is possible that the HDAC-inhibitory activity of sorafenib also results in the acetylation of RRM2, then reducing its protein stability.

The role of RRM2 in autophagy is rarely investigated. Only one previous study shows that RRM2 knockdown by siRNA and treatment with RR inhibitors can induce autophagy [55]. Although overexpression of RRM2 attenuates rapamycin-induced autophagy, the minimal effect of RRM2 overexpression alone on autophagy (LC3B expression) is found in that study [55]. In our study, we found that RRM2 overexpression did not alter sorafenib-induced autophagy in HCC cells, suggesting that sorafenib induces autophagy independent of RRM2 inhibition. In addition, RRM2 overexpression alone reduced LC3B-II expression in HepG2 cells, whereas the expression of LC3B-II was changed by RRM2 overexpression. Further investigation of the role of RRM2 in autophagy is needed.

In conclusion, this study identified that RRM2 is a novel target of sorafenib in HCC. Sorafenib inhibits RRM2 expression in both transcriptional and post-transcriptional manners. RRM2 inhibition by sorafenib was correlated with the autophagy-inducing and growth-inhibiting effects of sorafenib in HCC cells. However, ectopic RRM2 overexpression did not rescue HCC cells from the sorafenib-induced autophagy and growth inhibition, which may be explained by the observation that sorafenib efficiently reduces the ectopic RRM2 protein expression. Interestingly, long-term colony formation assay shows that RRM2 overexpression partially rescues HCC cells from sorafenib’s cytotoxicity, suggesting that RRM2 inhibition may also contribute to the anticancer activity of sorafenib in HCC. Nevertheless, the exact functional outcomes of RRM2 inhibition by sorafenib warrant further investigations.

## Figures and Tables

**Figure 1 biomolecules-10-00117-f001:**
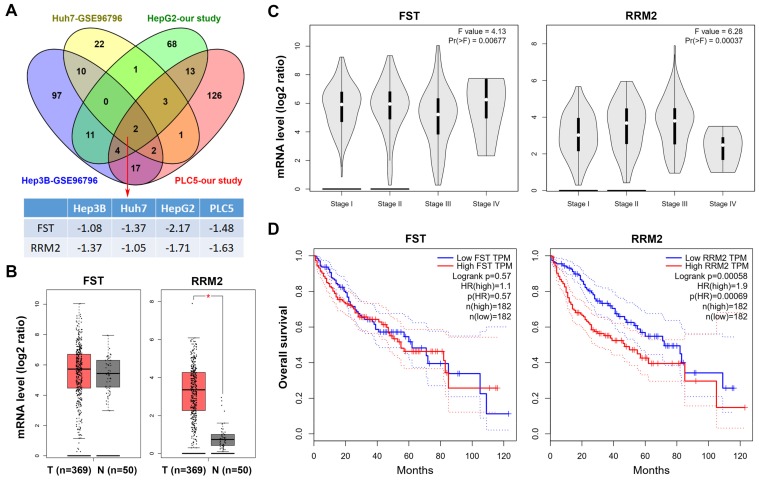
Role of follistatin (FST) and ribonucleotide reductase regulatory subunit M2 (RRM2) genes in hepatocellular carcinoma (HCC). (**A**) The differentially expressed genes (DEGs) in sorafenib-treated HCC cells obtained from our study (HepG2 and PLC5 cells) and the NCBI-GEO database (Huh7 and Hep3B cells; GSE96796). The overlapping genes are shown in a Venn diagram. The fold changes for FST and RRM2 genes in these data sets are shown in Appendix A. (**B**) FST and RRM2 mRNA expressions in tumor and normal tissues from HCC patients were analyzed using the GEPIA web tool. (**C**) FST and RRM2 mRNA expressions in tumor tissues from HCC patients in different stages were analyzed using the GEPIA web tool. (**D**) The impacts of FST and RRM2 mRNAs on the overall survival of HCC patients were analyzed using the GEPIA web tool.

**Figure 2 biomolecules-10-00117-f002:**
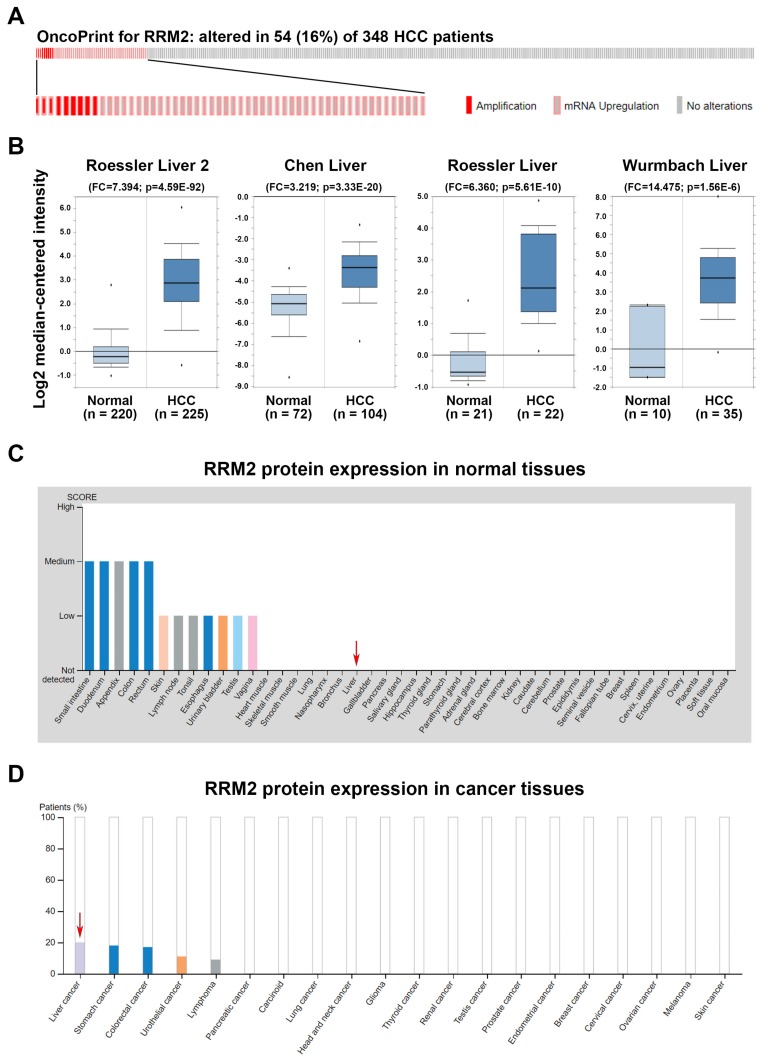
Ribonucleotide reductase regulatory subunit M2 (RRM2) expression in hepatocellular carcinoma (HCC). (**A**) A bar code plot (OncoPrint) for mutation status, copy number alterations, and mRNA expression of RRM2 gene in HCC were analyzed using the cBioPortal cancer genomics database. (**B**) RRM2 mRNA expression in tumor and normal tissues from HCC patients was analyzed using the Oncomine database. (**C**,**D**) RRM2 protein expression in normal (**C**) and tumor (**D**) liver tissues (highlighted by red arrows). Image credit: Human Protein Atlas. Data summary images were obtained from v19.proteinatlas.org, via the following links: https://www.proteinatlas.org/ ENSG00000171848-RRM2/tissue for (**C**) and https://www.proteinatlas.org/ENSG 00000171848-RRM2/ pathology for (**D**).

**Figure 3 biomolecules-10-00117-f003:**
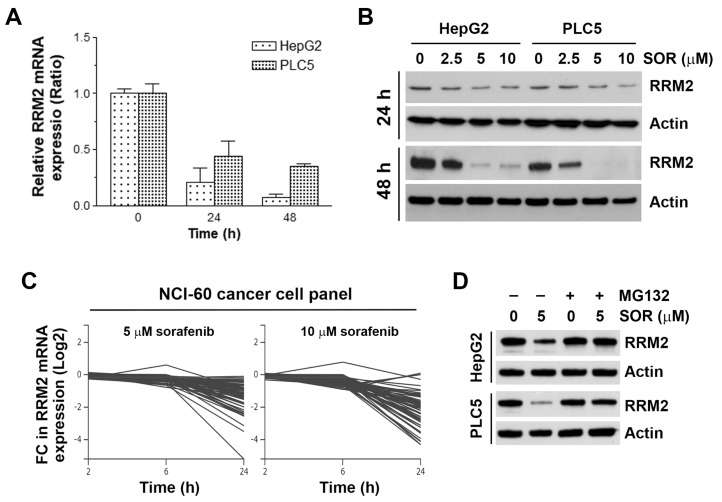
Effect of sorafenib on ribonucleotide reductase regulatory subunit M2 (RRM2) expression in hepatocellular carcinoma (HCC) cells. (**A**) HepG2 and PLC5 cells were treated with 5 μM sorafenib for 24 and 48 h, and then total RNAs were analyzed by real-time qPCR for RRM2 mRNA expression. (**B**) HepG2 and PLC5 cells were treated with 2.5, 5, and 10 μM sorafenib (SOR) for 24 and 48 h, and then RRM2 protein expression was analyzed by Western blot analysis. (**C**) The effect of sorafenib (5 and 10 μM sorafenib for 2, 6, and 24 h) on RRM2 mRNA expression (Log_2_ fold change) in a National Cancer Institute (NCI)-60 cancer cell panel was obtained from the NCI Transcriptional Pharmacodynamics Workbench database. (**D**) HepG2 and PLC5 cells were treated with 5 μM sorafenib (SOR) for 48 h. MG132, 5 μM, was added to cell culture 24 h before cell harvest for Western blot analysis.

**Figure 4 biomolecules-10-00117-f004:**
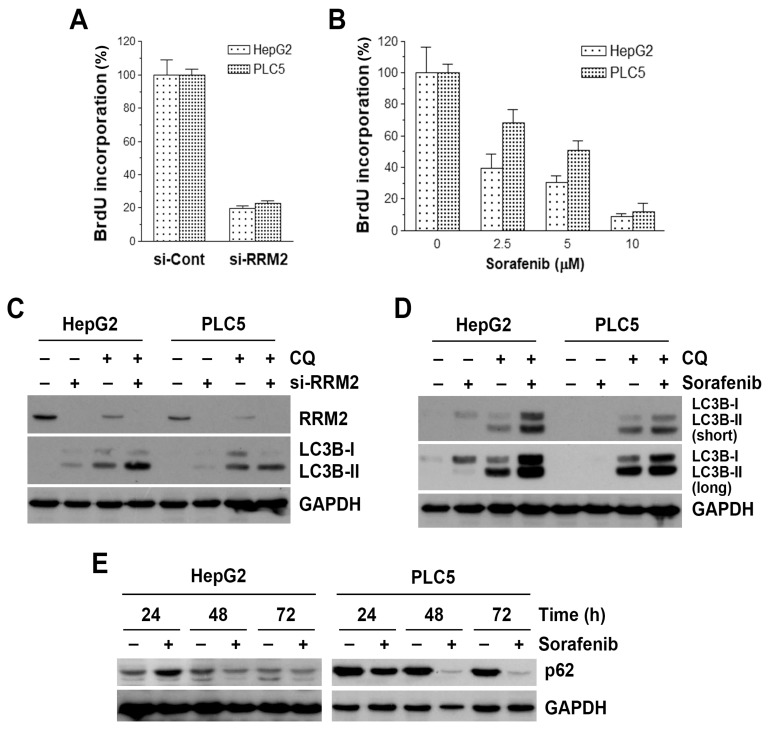
Effect of ribonucleotide reductase regulatory subunit M2 (RRM2) knockdown and sorafenib on cell proliferation and autophagy in hepatocellular carcinoma (HCC) cells. (**A**) HepG2 and PLC5 cells were transfected with RRM2 siRNA (si-RRM2) or the non-targeting control siRNA (si-Cont) for 72 h, and then BrdU cell proliferation assay was performed. (**B**) HepG2 and PLC5 cells were treated with 2.5, 5, and 10 μM sorafenib for 48 h, and then BrdU cell proliferation assay was performed. (**C**) HepG2 and PLC5 cells were transfected with RRM2 siRNA (si-RRM2) or the non-targeting control siRNA for 48 h. Chloroquine (CQ), 30 μM, was added to cell culture 6 h before cell harvest for Western blot analysis. (**D**) HepG2 and PLC5 cells were treated with 5 μM sorafenib for 48 h. CQ, 30 μM, was added to cell culture 6 h before cell harvest for Western blot analysis. (**E**) HepG2 and PLC5 cells were treated with 5 μM sorafenib for 24, 48, and 72 h, and then cells were harvest for Western blot analysis.

**Figure 5 biomolecules-10-00117-f005:**
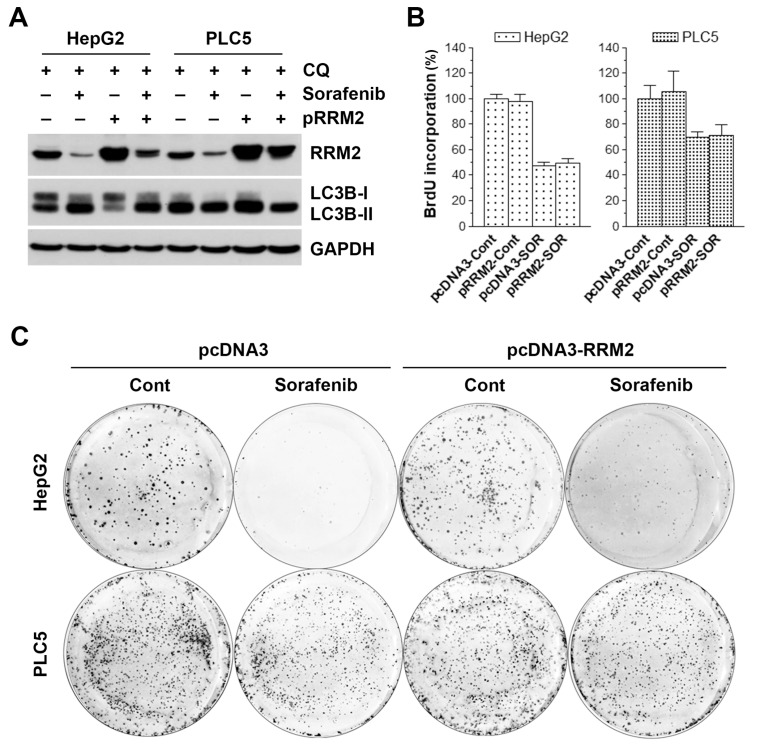
Effect of ribonucleotide reductase regulatory subunit M2 (RRM2) overexpression on sorafenib-induced cell death in hepatocellular carcinoma (HCC) cells. HepG2 and PLC5 cells were transfected with RRM2-encoding (pRRM2 or pcDNA3-RRM2) or control plasmid (pcDNA3). After 24 h, cells were treated with 5 μM sorafenib (SOR) for 48 h for Western blot analysis (**A**), BrdU cell proliferation assay (**B**), and colony formation assay (**C**). In (**A**), 30 μM chloroquine (CQ) was added to cell culture 6 h before cell harvest. In (**C**), cells were washed twice with PBS after sorafenib treatment. Cells were further cultured for an additional 10 to 14 days for colony formation.

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
