# Peer review of "Sorafenib Inhibits Ribonucleotide Reductase Regulatory Subunit M2 (RRM2) in Hepatocellular Carcinoma Cells"

_biomolecules, 2020, doi:10.3390/biom10010117_

Round 1

Reviewer 1 Report

Reviewer comments:

Using cancer genomics data, the authors identified that ribonucleotide reductase regulatory subunit M2 (RRM2) could be used as a potential prognosis biomarker as well as a therapeutic target for liver cancer (HCC).  

They also reported that RRM2 is a novel molecular target of sorafenib in HCC cell lines.

General comments:

There is significance and importance of this study especially in HCC patients. Liver cancer is much more common in sub-Saharan Africa and Southeast Asia. In some countries, it is the most common cancer type.

Identification of RRM2 as a novel target of sorafenib in HCC is the strength of this study.

The following concerns need to be addressed:  

Major concerns:

A: Based on their experimental evidences provided in Figure 4, the authors concluded that “Inhibition of RRM2 Induces Autophagy in HCC cells.” It is difficult to conclude the induction of autophagy only on the basis of LC3B western blot data. The authors mentioned in the same section (3.4) that accumulation of LC3B-II also indicated the inhibition of autophagy flux. So, they need to provide the results of other autophagy markers, like either induction of ATG5/ATG7/Beclin 1 etc. or inhibition of SQSTM1 (p62) to establish their conclusion.

B: Literature indicated that sorafenib induces autophagy, but cells treated with only sorafenib did not show any accumulation of LC3B-II in short exposure blot (Fig. 4D). Even in long exposure, only HepG2 cells showed negligible expression of LC3B-II (Fig. 4D, lane 2) and LC3B (both -I and -II) was undetected in PLC5 cells (Fig. 4D, lane 6). In this study, detection of other autophagy marker can only possibly support their conclusion that sorafenib induces autophagy.

C: Figure 4C clearly indicated that inhibition of autophagy by CQ drastically reduced the expression of RRM2 (lane 3 and lane 7) in both the cell lines. That means, possibly, the inhibition of autophagy decreases the expression of RRM2 in both HCC cell lines. But authors stated that “…..sorafenib induced autophagy …….possibly associated with the downregulation of RRM2. It is quite difficult to support this conclusion from the present results. More investigation is required to support this statement.

A: It is difficult to understand why the authors treated all of the experimental conditions with CQ that are presented in Figure 5A. I think this is not the best way to interpret the induction of autophagy.

B: In Figure 5A, untreated cell, the important control is missing. As mentioned before, result in Figure 4C clearly shows massive reduction of RRM2 by CQ but it was highly expressed in presence of CQ in Figure 5A. The authors need to explain.    

C: In HepG2 cells, LC3B-II drastically reduced after overexpression of RRM2 with CQ (Fig. 5A, lane 3, middle blot) whereas, in PLC5 cells, same LC3B-II expression was induced after overexpression of RRM2 with CQ (Fig. 5A, lane 7, middle blot). Why?

Minor comment:

In the title of Table 2 and in result section (3.1), it was mentioned about Huh7 and Hep3B cells. But, it the table, it was mentioned HepG2 and PLC5. Which one is correct?

Reviewer 2 Report

Yang et al. submitted a manuscript showing that RRM2 is partly responsible for the cytotoxic effects of sorafenib on hepatocellular carcinoma cells. However, I have the impression that the results presented do not confirm the conclusion contained in the title of the publication.

As shown in Figure 4, silencing of RRM2 induces cell autophagy. Furthermore, the cytotoxic effect of sorafenib is dependent on autophagy induction and correlates with a decrease in RRM2 expression. However, as shown on Figure 5 and 6, RRM2 overexpression did not inhibit sorafenib-induced autophagy, as well as proliferation. BrdU assay showed no significance difference between wild-type sorafenib-treated cells and sorafenib-treated RRM2 overexpressed cells. Importantly, RRM2 overexpression did not reduce MAP-LC3 level, thus autophagy induction and proliferation inhibition after sorafenib treatment, are not modulated by the elevated level of RRM2 protein.

From the results presented in the manuscript, it can be concluded that the action of sorafenib is not likely to depend on the presence of RRM2 in the cell.

It is shown that the presence of RRM2 is important for cells, because its silencing induced autophagy process and reduced cell proliferation. But whether sorafenib activity depends on RRM2 is not obvious from the results presented in the paper.  

Reviewer 3 Report

General comments

Sorafenib is a multi-kinase inhibitor with anti-proliferative and anti-angiogenic effects. It has been used as standard of treatment for advanced HCC. Although this study is of interest, it is inconsistently written. For example the introduction fails to smoothly lead the reader into the rationale and aims of the study. Moreover, it contains some misconceptions regarding HCC incidence (please see CA CANCER J CLIN 2018; 68: 394–424). This section may be re-written by adopting the following 4-paragraph format:

Define what liver cancer and HCC are. Write about HCC incidence around the world. Some concepts about HCC staging systems may be necessary. Establish connections between each HCC grade and treatment options. Use recent references. Define what sorafenib is. Show the pathways modulated by sorafenib. Define high throughput technologies and their role in determinate new HCC related pathways. There are recent high quality publish papers about it. State what these authors expected to find and why.

Rationale and design of the study are poorly elucidated. The methods must be described in same order of results. It is lacking the description of statistical analysis.

The first paragraph of discussion is important to summarize the gap in understanding that the manuscript is attempting to fill. So, the authors must re-write it. The authors discussed the findings in the setting of previously published articles, but it is necessary state the strengths and limitations of the present study.

Some specific comments and questions

I recommend do not use MoA as abbreviation for mechanism of action (Introduction section, last paragraph and results section – first paragrafh)

The sorafenib tosylate is an oral delivery form of the drug in vivo. Why did the authors choose this form of sorafenib? (Materials and Methods section – chemicals)

Why did not the authors perform MTT assay (short term viability assay)? (Material and methods section).

Real-time quantitative instead Realtime (Material and methods and results section).

What were the primers references? (Material and methods section).

The RNA-seq methodology needs better described with references.

Results, first paragraph: Thus, the understanding of mechanisms of action of sorafenib....instead Thus, more understanding of sorafenib´s MoAs...

The cells were treated with 5 µM. Why was used this concentration? The authors did not show data regarding cytotoxicity test or IC50.

Results section, first paragraph, line 10: analysed insted mined.

Tables 1 and 2: What was the fold change of differentially expressed genes? These genes may be classified according their molecular pathways.

Results section, pag 6, first line of item 3.2: To explore the role of…instead To gain more insights into.

Figure 4A. The authors state that inhibition of RRM2 expression by siRNA reduced cell viability. But the result shown is about cell proliferation.

Round 2

Reviewer 1 Report

Satisfactory.

Reviewer 3 Report

The authors were very kind to accept my suggestions. I am satisfied with the manuscript and I have no further comments.